# The Effectiveness of Calcium Phosphates in the Treatment of Dentinal Hypersensitivity: A Systematic Review

**DOI:** 10.3390/bioengineering10040447

**Published:** 2023-04-06

**Authors:** Mélanie Maillard, Octave Nadile Bandiaky, Suzanne Maunoury, Charles Alliot, Brigitte Alliot-Licht, Samuel Serisier, Emmanuelle Renard

**Affiliations:** 1Faculté de Chirurgie Dentaire, CHU Nantes, Service Odontologie Conser-Vatrice et Pediatrique, Nantes Université, F-44000 Nantes, France; 2Oniris, CHU Nantes, INSERM, Regenerative Medicine and Skeleton, RMeS, Nantes Université, UMR 1229, F-44000 Nantes, France; 3Faculté de Chirurgie Dentaire, CHU Nantes, Service Odontologie Restauratrice et Chirurgicale, Nantes Université, F-44000 Nantes, France

**Keywords:** dentin hypersensitivity, desensitizing agents, calcium phosphate, hydroxyapatite, nano-hydroxyapatite

## Abstract

Dentin hypersensitivity (DH) pain is a persistent clinical problem, which is a common condition known to affect patients’ quality of life (QoL), but no treatment has ever been agreed upon. Calcium phosphates, available in different forms, have properties that allow sealing the dentinal tubules, which may relieve dentin hypersensitivity. The aim of this systematic review is to evaluate the ability of different formulations of calcium phosphate to reduce dentin hypersensitivity pain level in clinical studies. The inclusion criterion was as follows: clinical randomized controlled studies using calcium phosphates in treating dentin hypersensitivity. In December 2022, three electronic databases (Pubmed, Cochrane and Embase) were searched. The search strategy was performed according to Preferred Reporting Items for Systematic Reviews and Meta-Analyses (PRISMA) guidelines. The bias assessment risks results were carried out using the Cochrane Collaboration tool. A total of 20 articles were included and analyzed in this systematic review. The results show that calcium phosphates have properties that reduce DH-associated pain. Data compilation showed a statistically significant difference in DH pain level between T0 and 4 weeks. This VAS level reduction is estimated at about −2.5 compared to the initial level. The biomimetic and non-toxic characteristics of these materials make them a major asset in treating dentin hypersensitivity.

## 1. Introduction

Dentin hypersensitivity (DH) is an oral complaint frequently reported in clinical dental practice. It is characterized by a short, sharp pain arising from exposed dentin in response to thermal, evaporative, tactile, osmotic, or chemical stimuli that cannot be ascribed to any other form of dental defect or pathology [1,2]. A review outlined a prevalence of DH ranging from 1 to 34% after clinical examination; the highest level has been reported to be on the cervical surface of the canine as well as first premolar permanent teeth and also in patients with periodontal alterations [3]. In their daily life, patients with dentin hypersensitivity complain of discomfort and pain while consuming hot or cold foods and beverages (coffee and ice cream) while toothbrushing or sometimes even while breathing. These symptoms and problems may be highly relevant, leading to restrictions on everyday activities and be a determinant of the individual’s oral-health-related quality of life (OHRQoL) [4].

Several theories have been proposed in order to explain the biological mechanism of dentin hypersensitivity [3,5,6]. To date, the most widely accepted theory of DH is the hydrodynamic theory of Brännström [7,8,9]. This theory is based on a rapid movement of the dentinal fluid after external stimuli, which indirectly activates the nociceptors contained in the interface of the pulp and dentine, triggering painful sensations [9]. This would explain why treatments that occlude dentinal tubules and reduce intratubular fluids movement showed beneficial effects with high to moderate certainty [10].

Many active principles have been tested for the treatment of dentin hypersensitivity, including desensitizing toothpastes, gels, varnishes, and mouth rinses. Numerous systematic reviews exist on this topic, and the results are sometimes conflicting. These products typically contain one or more active ingredients that work by modifying the nervous response, preventing or reducing the transmission of pain signals, and/or occluding the permeable dentinal tubules [11]. Water-soluble potassium salts such as potassium fluoride, potassium chloride, and, the most commonly used, potassium nitrate are active ingredients that reduce dentin hypersensitivity pain by decreasing the nervous excitability by depolarizing nervous cells in the dentin tubules [12], resulting in a decrease in the nerve excitability. Another active principle tested for dentin hypersensitivity is fluoride under different molecule forms: sodium fluoride, silver diamine fluoride, tin fluoride, and amine fluoride. These fluorides work by creating a physical barrier by precipating in the dentin surface and making it more resistant to acid erosion and other types of damage [13]. The oxalates are esters of oxalic acid, which can lead to the formation of calcium oxalate crystals by reacting to calcium ions from the oral cavity and occluding the dentinal tubules [14]. Arginine is an amino acid naturally found in saliva, able to blend with calcium carbonate and precipitate in dentinal tubules, resulting in the creation of a barrier resistant to acid dissolution [13]. Strontium also acts through the precipitation of particles on the exposed dentin and forming a protective barrier [15]. Other active ingredients such as sodium calcium phosphosilicate amorphous [16,17] promote the formation of apatite hydroxycarbonate on the dentin surface, occluding the dentinal tubules. Calcium phosphate, including nano-hydroxyapatite, can help to rebuild and strengthen the tooth structure by providing essential minerals that are lost during the demineralization process [18]. Physical agents such as glass ionomer, resins, and sealants are used in order to seal dentinal tubules and prevent the hydrodynamic dental pulp stimulation [19]. Glutaraldehyde is a molecule that reacts with serum albumin contained in dentinal fluid and is able to reduce the diameter of dentin tubules [19]. High-intensity lasers such as Nd:YAG, Er:YAG, Er, Cr:YSGG, and CO_2_ have been tested to reduce DH pain through the obliteration of dentinal tubules, whereas low-intensity lasers such as GaAIA or He-Ne may reduce DH pain symptoms by interfering with the Na^+^K^+^ ion pump in the cell membrane, in blocking the transmission of nerve stimulation [20,21]. Overall, the active principle being tested in dentin hypersensitivity depends on the specific product being used and the mechanism of action of the active ingredient. However, the goal of all these active principles is to provide relief from the discomfort associated with DH by reducing nerve sensitivity, remineralizing the tooth surface, and providing a protective barrier over the exposed dentin. All these procedures are considered as therapeutic treatments and can be delivered either in-home or in-office. A systematic review comparing the effectiveness of DH treatment showed that dentinal tubules occlusion as well as nerve desensitisation in the at-home or in-office conditions of delivery had similar effects [22]. This multitude of treatments is able to decrease the patient’s DH, but none of them constituted a gold-standard agent.

Since the 1950s, ceramic hydroxyapatite (HA) granules for bone defect repair have been reported [23], and in late 1980s, the first self-hardening calcium phosphate cements (CPC) were developed [24]. Indeed, as explained by Chow, 2009 [24], calcium phosphate cement containing an adequate concentration of tetracalcium phosphate and dicalcium phosphate anhydrous has a very high solubility, which enables precipitation in HA, a molecule whose general formula is Ca_10_(PO_4_)6(OH)_2_, which is highly biocompatible and has low solubility. HA is widely applied in medicine and dentistry as a bone substitute [24,25,26].

Hydroxyapatite is already used as a DH desensitizer [27]. Other molecules similar to HA, such as synthetic nano-Hydroxyapatite (n-HA) or soluble molecules able to self-set to a hard mass under HA form, such as Tetracalcium phosphate (TTCP) and dicalcium phosphate dihydrate (DCPD) [28], have already been tested in clinical conditions in the treatment of DH. These studies show encouraging results. However, no systematic reviews evaluated the effects of these molecules on the DH pain level.

This systemic review aims to evaluate the effect of various calcium phosphate molecules such as hydroxyapatite, nano-hydroxyapatite, TTCP, DCPD, dicalcium phosphate anhydrous (DCPA), or/and amorphous calcium phosphate (ACP) on the reduction in DH pain level.

## 2. Materials and Methods

The study protocol was registered in the International Prospective Register of Systematic Reviews (PROSPERO) under ID = CRD42022336712. The present systematic review was conducted per the Preferred Reporting Items for Systematic Reviews and Meta-Analyses (PRISMA 2020) 2020 guidelines [29]. Population, Intervention, Comparison, Outcomes, and Study design components of this systematic review are as follows: Participants (P) were adult patients suffering from dentin hypersensitivity due to non-carious cervical lesions and not associated with post-bleaching hypersensitivity and periodontal therapy. Interventions (I) were in-office or in-home treatments of dentin hypersensitivity with products containing calcium phosphate. For Comparison (C), the comparison with other molecules is not applicable, but we looked at the variation in the level of pain felt by the patients before and after treatment with calcium phosphates. Outcome (O) was the reduction in pain associated with dentin hypersensitivity after treatment with calcium phosphate molecules. The study design (S) selected was a randomized controlled trial (RCT). Case reports, in vitro studies, in situ studies, systematic reviews, meta-analysis, letters to editors, and non-randomized trials, as well as studies on tooth decay or studies with no good molecule tested, were excluded. The research question was as follows: Are calcium phosphate able to reduce the DH pain?

### 2.1. Search Strategy

Three databases (PubMed/Medline, Cochrane Library, and EMBASE) were searched using relevant keywords to identify articles published until December 2022, with no language restriction, as shown in Table 1. Additionally, bibliographies of all selected articles, specialized journals, and other related publications, including reviews and meta-analyses, were also searched to identify further relevant articles. The records obtained from this extensive literature search were transferred to an EndNote^®^ library, and duplicates were removed.

### 2.2. Screening and Study Selection

The research and selection process articles were carried out independently by two authors (M.M. and S.M.). First, the retrieved articles were imported into a bibliographic reference management software program (EndNote), where duplicates were removed. Then, the records’ titles and abstracts obtained were screened, based on determined eligibility criteria. Finally, the full texts of the remaining studies were assessed by the same authors. Discrepancies were resolved, and consensus was built by engaging a third author (E.R.). Only randomized controlled trials that assessed the dentinal desensitized effect of calcium phosphate were included.

### 2.3. Data Extraction

When available, the data of included studies were extracted by both reviewers (M.M and S.M) and verified and confirmed by two other authors (O.N.B. and E.R). An Excel file was previously established to provide support for collecting demographic data (name of first author, year and country of publication, number of participants, and mean age), study methodology (study design, number and characteristics of the participating groups, number of follow-up visits, method of measuring dentin hypersensitivity, composition, concentration and use of calcium phosphate as a desensitizing agent for dentine hypersensitivity), and main results. All these extracted data were listed in Table 2.

### 2.4. Quality Assessment

The same two authors (M.M. and S.M.) assessed the risk of bias in the included studies using Cochrane’s Collaboration tool for assessing the risk of bias in randomized controlled trials [50]. Disagreements were resolved via discussion, and a third researcher (E.R.) was approached when necessary. This evaluation concerned the generation of the randomization sequence (selection bias), concealment of the allocation (reporting bias), blinding of the investigator and the participant (confusion bias), blind evaluation of the results (performance bias), management of missing data (attrition bias), selection of the reporter, and other types of bias. From these criteria, the bias risk level was determined to be low, unclear, or high.

### 2.5. Synthesis of Results

A qualitative and quantitative synthesis of the results of the included studies, structured around different outcomes, was performed. The data from these different studies were extracted, and the results are summarized in Table 3. For studies in which the authors reported results as medians and interquartile ranges, the values were converted to means and SDs using the formula (q1 + median + q3)/3, where q1 indicates the 25th percentile and q3 the 75th percentile. An approximation of the standard deviation was obtained by applying this formula (q3 − q1)/1.35. Analysis groups between baseline and 4 weeks of follow-up were formed according to the method of assessment of dental hypersensitivity to determine whether calcium phosphates are effective in reducing pain associated with dental hypersensitivity associated pain. Data from these different groups were pooled to determine the mean pain reduction value. When studies used the same type of intervention and comparison groups with the same outcome measure, the results were pooled with mean differences for continuous outcomes.

## 3. Results

### 3.1. Study Selection

The initial search of all sources yielded 10,019 records. Of these, 2435 duplicated studies were removed using the reference manager EndNote^®^. A total of 7515 articles were excluded after reading titles and/or abstracts, 12 records were excluded since reports were not retrieved, 57 records from database registers and 18 identified through other methods were read and analyzed in their full-text, and 55 records were excluded for reasons such as not good drugs tested (*n* = 34), in vitro studies (*n* = 11), or study on tooth decay (*n* = 10), as shown in Figure 1. Twenty records met the inclusion criteria and were included in the systematic review: Poliakova et al., 2022 [30], Alharith et al., 2021 [31], Amaechi et al., 2021 [32], Eyuboglu et al., 2020 [33], Usai et al., 2019 [34], Amaechi et al., 2018 [35], Ameen et al., 2018 [36], Anand et al., 2018 [37], Vano et al., 2017 [38], De Oliveira et al., 2016 [39], Wang et al., 2016 [40], Gopinath et al., 2015 [41], Jena et al., 2015 [42], Mehta et al., 2015 [43], Naoum et al., 2015 [44], Mehta et al., 2014 [45], Porciani et al., 2014 [46], Vano et al., 2014 [47], Ghassemi et al., 2009 [48], and Geiger et al., 2003 [49]. The selection process has been detailed in the attached PRISMA flowchart (Figure 1).

### 3.2. Description of Included Studies

The characteristics of the 20 included articles are presented in Table 2. The number of subjects included varied from 8 to 208. The age range of patients ranged from 18 to 80 years. The follow-up range varied from immediately to 6 months. Most of the studies performed a 4-week follow-up phase [30,32,33,34,35,36,37,38,39,40,41,42,47,48,49].

Different formulations of calcium phosphate were used by authors: hydroxyapatite, nano-hydroxyapatite in different concentrations, amorphous calcium phosphate (ACP), tetracalcium phosphate (TTCP), dicalcium phosphate anhydrous (DCPA) and dicalcium phosphate dihydrate (DCPD), and tri-calcium phosphate (TCP). Molecules were tested in the form of toothpaste, gel, or chewing gum and administered through an in-office treatment in 10 studies [31,33,34,35,36,39,42,43,45,49]. In the other 10 studies, the treatment was performed by the patients themselves, at home [30,32,37,38,40,41,44,46,47,48]. Most of the included studies used another desensitizing agent as a control [30,31,32,33,34,35,36,37,38,39,40,41,42,44,45,48], and four studies used only a placebo as a control [43,46,47,49].

Dentin hypersensitivity is generally assessed through different tests. In all included studies, dentin hypersensitivity was evaluated with an air blast test, tactile test, or cool water test, in accordance with the guidelines described by Holland et al., 1997 [2]. These guidelines, recommending at least two tests, were respected by most of the studies, except for [30,40,48], which only used an air blast test. The majority of studies realized an air blast assessment (evaporative stimulus) associated with a tactile sensitivity test [31,33,34,36,38,41,42,43,44,45,46,47,49]. Three studies [36,41,46] also used cold tests or cold-water tests. Two studies [32,39] associated the air blast test with a cold test. Anand et al. [37] recorded the amperage value of an electric test. The dentin hypersensitivity pain was recorded with a visual analogic scale of 100 mm (VAS) [31,32,33,34,35,38,39,40,41,42,43,45,47,48,49] or a Schiff scale (SCASS) with a score from 0 to 3 [30,36,46], except for [37], which used an amperage value, and [44], which used an NRS-11 pain rating scale.

All studies showed significant reductions in VAS or Schiff of dentin. In order to determine the efficacy of calcium phosphate in the reduction in DH pain level, we synthetized data in Table 3 accordingly with the realized test. Data compilation showed a statistically significant difference in DH pain level between T0 and 4 weeks. This reduction is estimated at about −2.5 compared to the initial level of pain.

Nine studies showed a significant decrease in VAS or Schiff at 4 weeks for the air blast stimulation [30,33,34,38,39,40,41,45,47]. The total calculated mean difference score of all studies between the baseline T0 and 4 weeks was −2.71 ± 0.07 (−2.85 to −2.57) *p* < 0.05. Desensitized agents used were n-HA [34,39,48], n-HA15% [36,38,47], n-HA20% [30,40], n-HA 25% [36], and TTCP/DCPA [33,34,45]. All data are compiled in Table 3.

Six of seven studies showed a significant decrease in VAS or Schiff at 4 weeks for tactile stimulation [33,34,36,38,45,47]. The decrease was not significant in one study [41]. The total calculated mean difference score of the seven studies was −2.53 ± 0.07 (−2.66 to −2.39) *p* < 0.05. All data are compiled in Table 3.

Three studies showed a significant decrease in VAS at 4 weeks for the cold water test [36,39,41]. The total calculated mean difference score of all studies between the baseline T0 and 4 weeks was −2.56 ± 0.16 (−2.88 to −2.23) *p* < 0.05. Desensitized agents were n-HA. All data are compiled in Table 3.

### 3.3. Analysis of the Risks of Bias

The results of the risk of bias assessment are presented in Figure 2. This analysis was carried out using the Cochrane Collaboration tool [50]. This assessment involved randomized clinical trials and was carried out on all the studies included in this systematic review. The assessment revealed that seven studies were considered to have a low risk of bias [32,34,35,36,41,44,48]. Five studies were considered to be at high risk of bias for the following reasons: in the study of Eyuboglu et al. [33], because the randomization was performed after the initial pain assessment; in the study of Gopinath et al. [41], because the randomization method and the description of the sample size were not clearly exposed; and in the study of Jena et al. [42], because the absence of description of the sample size and of duration and location of the study also constituted a risk of bias.

## 4. Discussion

Recent systematic reviews with meta-analyses have compared the effectiveness of several desensitizing toothpaste formulations, including some containing nano-hydroxyapatite or potassium combined with hydroxyapatite [10,18,19,51,52,53]. In our review, we focused specifically on the effectiveness of hydroxyapatite and other calcium phosphate materials able to self-set to a hard mass [24], such as tetracalcium phosphate (TTCP) and dicalcium phosphate dihydrate (DCPD) powders, which are able to produce a supersaturated solution and faster hydroxyapatite precipitation due to their high solubility at neutral PH [28]. In this systematic review, we show the beneficial clinical effects of all different calcium phosphate formulations on dentin hypersensitivity. Whatever the test used (air blast, tactile or cold water), calcium phosphate induced a reduction in a mean of 2.5 pain level on the VAS scale after 4 weeks. Additionally, significant beneficial effects appeared immediately after treatment in eight studies [31,33,34,37,39,42,43,45].

These results may be explained by the ability of calcium phosphates to spontaneously form hydroxyapatite at physiological pH and to adhere to the exposed dentine, forming a layer of calcium phosphate components, which may allow them to seal exposed dentinal tubules and consequently be a good candidate for the treatment of dentin hypersensitivity with a VAS drop immediately after the application.

The beneficial effect, as described in our review, is in accordance with the results of a large systematic review and meta-analysis conducted by Marto et al., regarding numerous molecules in the treatment of DH [19]. In this review, hydroxyapatite and other calcium phosphate molecules showed a significant reduction in DH pain at different points in time.

A previous systematic review and meta-analysis conducted by de Melo Alencar et al. in 2019 [18] underlined the effectiveness of nano-hydroxyapatite in the relief of dentin hypersensitivity compared to n-HA free treatment. Indeed, Alencar et al. [18] showed a significant desensitizing effect against evaporative and tactile stimuli but not against cold stimulation. They hypothesized that cold stimulus, the most disturbing test, could involve not only the hydrodynamic theory but also other factors such as TRPM8 channels in odontoblasts. Concerning the air blast test and the tactile test, their results are in accordance with our systematic review, except for the cold water, for which we show a significant reduction in DH pain level after phosphate calcium treatment. De Melo Alencar et al. also compared n-HA to placebo or other desensitizing agents, particularly arginine. This amino acid when combined with calcium carbonate mimics saliva’s ability to occlude and seal open dentinal tubules, which renders that tooth surface resistant to acid and thermal attacks. It has been shown by two previous meta-analyses and one systematic review as promising bioactive agent [13,54,55] in DH. In their review, Alencar et al. [18] showed, in the 4-week follow-up, a better result with nano-hydroxyapatite than those presented by arginine in the treatment of dentin hypersensitivity. In our study, we did not compare calcium phosphate with other products or placebo, since this was not our research question.

However, another systematic review and meta-analysis conducted by Hu et al. [56], comparing numerous dentin desensitizing agents, showed a very low level of evidence of nano-hydroxyapatite and amorphous calcium phosphate toothpaste compared with other desensitizing agents. Indeed, this study included fifty-three clinical studies, but only four with calcium phosphate, two with nano-hydroxyapatite and two with amorphous calcium phosphate, which considerably reduces the effect of this evidence [56].

The review by Cunha-Cruz et al. [57] showed a significant effect of n-HA on DH pain levels in two studies but no effects from amorphous calcium phosphate, but this author took into account only results included in previous systematic reviews and meta-analyses [56].

It is worth noting that the hypersensitivity reduction efficacy of nano-hydroxyapatite was increased du to the adjunction of sodium [58] or ionometric sealant [59] or when combined with laser treatment [60].

Furthermore, the concentration of calcium phosphate used is probably an important factor for effectiveness. Shetty et al. [61], in an in vitro study, reported an enhanced desensitizing action with 100% nano-hydroxyapatite over 8 weeks of treatment compared to 25% nano-hydroxyapatite, and the authors concluded that increased concentrations of the molecule increased its penetration into the tubules and probably improved its desensitizing ability. Our study does not allow us to determine the suitable concentration to use to reach the most efficient effect.

Other biomaterials containing calcium such as calcium sodium phosphosilicate seem to have interest in the treatment of dentin hypersensitivity [30,46,47]. Finally, it is important to note that the hypersensitivity was not completely resolved regardless of the treatment applied. According to our results, a level of pain persists after 4 weeks of treatments as shown in Table 3, which could be explained by the fact that the effectiveness of the treatment tends to decrease or disappear over time [34,49].

Alharith et al. [31], De Oliveira et al. [39], Porciani et al. [46], and Geiger et al. [49] showed positive results in the placebo groups, resulting in a significant effect of treatment but no significant reduction compared to the placebo group. In [31,39,46,49], patients felt significant reductions of up to 60% of dentin hypersensitivity following the application of the placebo treatment. These positive effects of placebo treatments are important to consider, since there may be other factors that could explain the effectiveness of desensitizing therapeutics agents in the reduction in dentin hypersensitivity such as desensitizing agents in control groups. Moreover, these lower levels of sensitivity in placebo groups can also be attributed to the well-known Hawthorne effect, which describes the modification of behavior when individuals are aware that they are being observed, which could influence the patient’s responses and ca lead to bias in healthcare studies [62]. The placebo effect may also be involved, since positive motivation and emotional stimuli could activate pain inhibitors in the central nervous system [63].

Our systematic review presents limits that required results to be carefully interpreted. Indeed, included studies are still heterogenous in terms of the age of patient from 18 to 80 years, where we do not know if age is an influencing factor of DH level pain. Additionally, different desensitized agents in different concentrations and different application modalities were used by different authors, and the evaluation of the DH realized with three different tests could be different according to the team’s research; all these limits did not allow the generalization of results.

## 5. Conclusions

This systematic review shows a reduction in pain perception after calcium phosphate application immediately and 4 weeks after treatment, making this biomaterial a good candidate for the relief of dentin hypersensitivity.

## Figures and Tables

**Figure 1 bioengineering-10-00447-f001:**
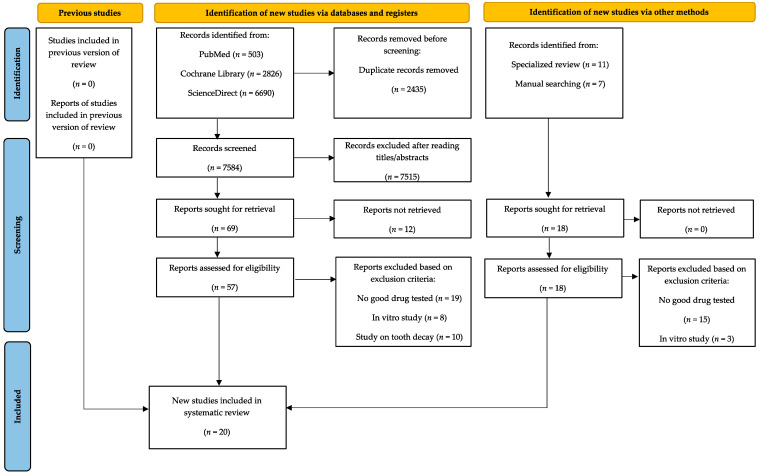
PRISMA flow diagram.

**Figure 2 bioengineering-10-00447-f002:**
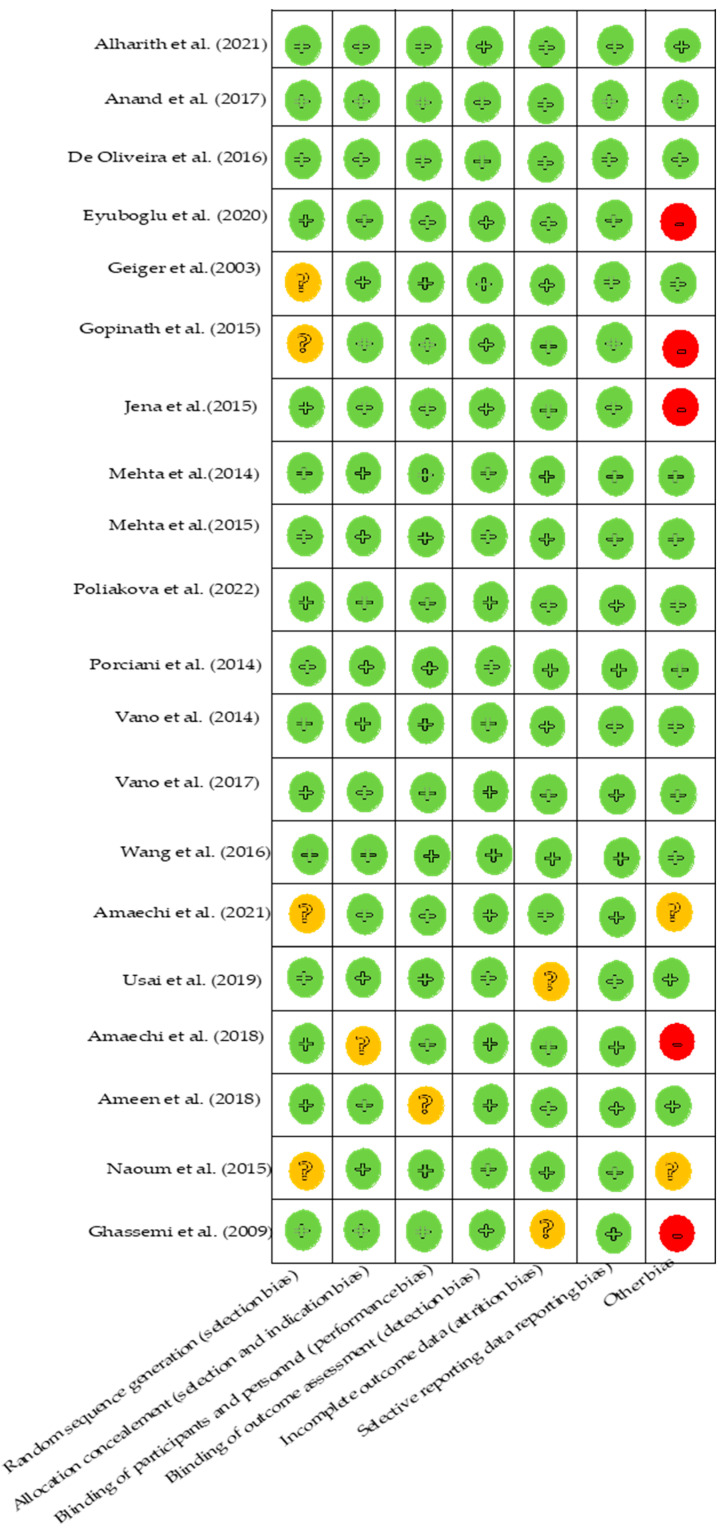
Risk of bias of included studies green indicates low riskof bias [30,31,37,38,39,40,43,45,46,47], orange indicates uncertain or moderate risk of bias [32,34,36,44,49], and red indicates high risk of bias [33,35,41,42,48].

**Table 1 bioengineering-10-00447-t001:** Database and search terms.

**Pubmed (filters applied: Randomized Control Trial, Human)**	(“Dentin Sensitivity” [Mesh] OR Sensitivities, Dentin OR Sensitivity, Dentin OR Dentine Hypersensitivity OR Dentine Hypersensitivities OR Hypersensitivities, Dentine OR Hypersensitivity, Dentine OR Dentine Sensitivity OR Dentine Sensitivities OR Sensitivities, Dentine OR Sensitivity, Dentine OR Tooth Sensitivity OR Sensitivities, Tooth OR Sensitivity, Tooth OR Tooth Sensitivities OR Dentin Hypersensitivity OR Dentin Hypersensitivities OR Hypersensitivities, Dentin OR Hypersensitivity, Dentin) AND (“Dentin Desensitizing Agents” [Mesh] OR Agents, Dentin Desensitizing OR Desensitizing Agents, Dentin OR “Tooth Remineralization” [Mesh]) AND (“Calcium Phosphates” [Mesh] OR dicalcium phosphate OR calcium monohydrogen phosphate dihydrate OR dicalcium phosphate dihydrate OR dibasic calcium phosphate dihydrate OR calcium phosphate, dihydrate OR calcium phosphate, dibasic OR dicalcium phosphate anhydrous OR brushite OR morphous calcium phosphate OR nHAC composite OR “Hydroxyapatites” [MeSH Terms] OR Hydroxyapatite Derivatives)
**Cochrane library (All text)**	Dentine Hypersensitivity OR Tooth Sensitivities OR Agents Dentin Desensitizing OR Remineralization tooth AND dicalcium phosphate OR calcium monohydrogen phosphate dihydrate OR dicalcium phosphate dihydrate OR dibasic calcium phosphate dihydrate OR calcium phosphates OR dihydrate calcium phosphate OR brushite OR Hydroxyapatite Derivatives OR Amorphous calcium phosphate OR nHAC composite
**Embase (filters applied: Human, controlled study)**	(‘dentine hypersensitivity’/exp OR ‘dentine hypersensitivity’ OR ((‘dentine’/exp OR dentine) AND (‘hypersensitivity’/exp OR hypersensitivity)) OR ‘tooth sensitivities’ OR ((‘tooth’/exp OR tooth) AND sensitivities) OR ‘agents dentin desensitizing’ OR (agents AND (‘dentin’/exp OR dentin) AND desensitizing) OR ‘remineralization tooth and dicalcium phosphate or calcium monohydrogen phosphate dihydrate‘ OR ((‘remineralization’/exp OR remineralization) AND tooth and dicalcium AND phosphate or AND calcium AND monohydrogen AND (‘phosphate’/exp OR phosphate) AND dihydrate) OR ‘dicalcium phosphate dihydrate‘ OR (dicalcium AND (‘phosphate’/exp OR phosphate) AND dihydrate) OR ‘dibasic calcium phosphate dihydrate‘ OR (dibasic AND (‘calcium’/exp OR calcium) AND (‘phosphate’/exp OR phosphate) AND dihydrate) OR ‘calcium phosphates’/exp OR ‘calcium phosphates’ OR ((‘calcium’/exp OR calcium) AND (‘phosphates’/exp OR phosphates)) OR ‘dihydrate calcium phosphate ‘ OR (dihydrate AND (‘calcium’/exp OR calcium) AND phosphate) OR ‘brushite’/exp OR brushite OR ‘hydroxyapatite derivatives’ OR ((‘hydroxyapatite’/exp OR hydroxyapatite) AND derivatives) OR ‘amorphous calcium phosphate‘ OR (amorphous AND (‘calcium’/exp OR calcium) AND phosphate) OR ‘nhac composite’ OR (nhac AND (‘composite’/exp OR composite))) AND ([controlled clinical trial]/lim OR [randomized controlled trial]/lim)

**Table 2 bioengineering-10-00447-t002:** Studies included in quantitative synthesis.

Author, Year, Country	Participants (ST)	Age Range Mean Age (SD)	Study Design	Study Group (*n*)	Evaluation Method	Results
**Poliakova et al., 2022, Russia [30]**	30 (NR)	35–45 years 37.5 (2)	RCT, DB	G1(*n* = 10): 20% n-HA paste; G2 (*n* = 10): nZnMgHA paste; G3 (*n* = 10): nFAP paste Toothbrushes twice daily for a month	Schiff Index values of CAS at baseline (T0) and after 2 (T1) and 4 (T2) weeks	At 4 weeks the Schiff Index score of 20% n-HA decreased significantly compared to baseline.
**Alharith et al., ** **2021, Saudi Arabie [31]**	63 (126)	18–60 years 39 (NR)	RCT, DB	G1 (*n* = 21): 15% n-HA paste; G2 (*n* = 21): fluoride paste; G3 (*n* = 21): placebo Single application of the paste at the baseline visit and 1 week follow up visit	VAS scores of TS and CAS evaluated at baseline (T0), immediately after paste application (T1), and after 1 week (T2)	A statistically significant reduction in VAS scores of TS and CAS tests from T0–T1 and T0–T2. Single application of n-HA paste significantly reduces DH.
**Amaechi et al., ** **2021, USA [32]**	85 (NR)	18–80 years 50.8 (11.4)	RCT, DB	G1 (*n* = 22): 10% n-HA; G2 (*n* = 19): 15% n-HA; G3 (*n* = 24): 10% n-HA + 5% Potassium Nitrate; G4 (*n* = 20): 15% SCPS Toothbrush during 2 min twice a day for 8 weeks	VAS scores of CAS and cold test at baseline (T0), 2 weeks (T1), 4 weeks (T2), 6 weeks (T3), and 8 weeks (T4).	All concentrations of n-HA showed a significant decrease in VAS scores at each time point.
**Eyuboglu et al., ** **2020, Turkey [33]**	40 (121)	18–65 years 41.35 (NR)	RCT, DB	G1 (*n* = 10; *n* * = 21): TTCP/DCPA; G2 (*n* = 10; *n* * = 36): Sodium Fluoride 5%, TCP Xylitol; G3 (*n* = 10; *n* * = 33): SR Monomer Matrix + 2-hydroxyethyl methacrylate; G4 (*n* = 10; *n* * = 31): 2-hydroxyethyl methacrylate + glutaraldehyde. One application according to manufacturer’s instruction	VAS score of TS and CAS evaluated at baseline (T0), immediately after application (T1), after 1 day (T2), after 2 weeks (T3) and after 4 weeks (T4)	TTCP/DCPA showed a significant difference between T0 and T1 and a significantly lower CAS score at T4 than T2 and T3.
**Usai et al., ** **2019, Italy [34]**	105 (210–420)	20–50 years 43–50 (NR)	RCT, DB	G1 (*n* = 35): TTCP/DCPA 30 s application; G2 (*n* = 35): DD (premixed n-HAP alcohol-based gel) 45 s application; G3 (*n* = 35): BWE (premixed n-HAP water-based gel) 10 min application	VAS scores of TS and CAS were recorded at baseline (T0), immediately after (T1), at 1 week (T2), 4 weeks (T2), 12 weeks (T3) and 24 weeks (T4).	TTCP/DCPA paste showed a statistically significant decrease in DH after 24 weeks in comparison to T0.
**Amaechi et al., ** **2018, USA [35]**	50 (NR)	18–80 years 45.47 (13)	RCT, DB	G1 (*n* = 25): 20%n-HA; G2 (*n* = 25): 20% Silica. One application during 5 min after the 2 min before-bed brushing teeth and water rinsing.	VAS scores of CAS and cold stimulation at baseline (T0), 2 weeks (T1), 4 weeks (T2), 6 weeks (T3) and 8 weeks (T4)	VAS score of CAS indicated significant reduction in DH at each time point with either n-HAP.
**Ameen et al., ** **2018, Egypt [36]**	10 (40)	20–45 years (NR)	RCT, DB	G1 (*n* * = 10): 15%n-HA + 1%NaF; G2 (*n* * = 10): 15%n-HA; G3 (*n* * = 10): 25%n-HA + 1%NaF; G4 (*n* * = 10): 25%n-HA. Four applications during 1 min at T0, 1 day, 1 week, 2 weeks and 4 weeks	Schiff index values for TS, CAS and Cold stimulation were evaluated at baseline (T0), 1 day (T1), 1 week (T2), 2 weeks (T3) and 4 weeks (T4)	n-HA molecules showed significant effects on DH at T1 compared to T0. The level of the Schiff index was back to 0 for all groups after 2 weeks.
**Anand et al., ** **2018, India [37]**	60 (60)	18–50 years 42.33 (7.58)	RCT, DB	G1 (*n* = 30): 8% arginine paste; G2 (*n* = 30): 1% n-HA paste. One application of 1 cm of toothpaste directly to the sensitive site of the selected tooth, then brushing for two minutes twice a day for 4 weeks	Amperage values were recorded at baseline, 5 min, 1 and 4 weeks with Digitest II (PARKELL, Inc., New York, NY. USA)	n-HA containing toothpastes provided a statistically significant reduction in DH, 5 min, 1 and 4 weeks after application.
**Vano et al., ** **2017, Italy [38]**	105 (NR)	20–70 years (NR)	RCT, DB	G1 (*n* = 35): 2% n-HA gel paste; G2 (*n* = 35): fluoride gel paste; G3 (*n* = 35): placebo, 10 min application twice a day during 4 weeks	VAS scores for TS and CAS were evaluated at baseline (T0) and after 2(T1) and 4(T2) weeks	n-HA in gel toothpaste significantly reduced the DH between baseline and 4 weeks
**De Oliveira et al., ** **2016, Brazil [39]**	8 (138)	22–48 years 29.5 (NR)	RCT, DB	G1 (*n* * = 33): Strontium acetate/calcium carbonate 60 s application; G2 (*n* * = 31): Calcium Carbonate/8% Arginine 3 s application by repeating 1 time the procedure; G3 (*n* * = 39): n-HA 10 s application with rest of 5 min; G4 (*n* * = 35): toothpaste without fluoride, 60 s application	VAS scores of CAS and cold stimulus with tetrafluoroethane were evaluated at baseline (T0), immediately after paste application (T1), after 24 h (T2) and after 30 days (T3).	n-HA showed significant difference for CAS and cold test between T0 and T3 days.
**Wang et al., ** **2016, Brazil [40]**	28 (137)	18–60 years (NR)	RCT, DB	G1 (*n* * = 31): 20%n-HA paste + NaF; 9000 ppm F; G2 (*n* * = 22): 20%n-HA + home-care pastes (10% HA, potassium nitrate, and NaF; 900 ppm F); G3 (*n* * = 28): 8% arginine + home-care toothpaste (8% arginine, sodium monofluorophosphate, 1450 ppm F); G4 (*n* * = 45): Duraphat. One application twice a day after toot brushing for 3 months	VAS scores of CAS was evaluated at baseline (T0) and after 1(T1) month and 3(T2) months	n-HA toothpaste was effective treatment for reducing DH over three months.
**Gopinath et al., ** **2015, India [41]**	36 (NR)	18–60 years (NR)	RCT, DB	G1 (*n* = 18): n-HA paste; G2 (*n* = 18): 5% SCPS. Brushing for two minutes and no more than twice a day in total during 4 weeks	VAS scores of TS, CAS and cold water tests were recorded at baseline (T0) and after 4 weeks (T1).	n-HA paste showed significant reduction in DH after 4 weeks
**Jena et al., ** **2015, India [42]**	45 (122)	18–50 years (NR)	RCT, DB	G1 (*n* * = 40): 5% Novamin paste; G2 (*n* * = 40): 8% arginine paste; G3 (*n* * = 42): 15% n-HA paste, 60 s application	VAS scores of TS and CAS at baseline (T0) immediately (T1), 1 (T2) and 4 (T3) weeks after treatment	n-HA showed significant reduction in VAS immediately, after 1 and 4 weeks.
**Mehta et al., ** **2015, India [43]**	35 (70)	18–42 years 33.3 (NR)	RCT, DB	G1 (*n* * = 35): TTCP/DCPA; G2 (*n* * = 35): Placebo, 30 s application	VAS scores to TS and CAS at baseline T0, 15 min after treatment (T1), 1 day (T2), 1 week (T3), 3 (T4) and 6 (T5) months	TTCP/DCPA toothpaste show a decrease in DH progressively from (T1) to (T5).
**Naoum et al., ** **2015, Australia [44]**	71 (NR)	39–45 years (NR)	RCT, DB	G1 (*n* = 20): Colgate Cavity Protection (1000 ppmF-MFP); G2 (*n* = 17): Sensodyne Total Care (1000 ppmF-NaF + 19,300 ppmK+ -KNO3); G3 (*n* = 16): Clinpro Tooth Creme (950 ppmF-NaF + f TCP); G4 (*n* = 18): Clinpro Tooth Creme (brushing + additional topical application). Toothbrush twice daily for 10 weeks	NRS-11 pain rating scale of TS, CAS and hypertonic solution were assessed at baseline (T0), 6 weeks (T1), and 10 weeks (T2)	fTCP (brushing + additional topical application) showed a significant reduction in DH.
**Mehta et al., ** **2014, India [45]**	49 (200)	18–50 years (NR)	RCT, DB	G1 (*n* * = 50): MSC 30 s application; G2 (*n* * = 50): NAN 20 s application; G3 (*n* * = 50): TTCP/DCPA (TMD) 30 s application; G4 (*n* * = 50): GLU, 60 s application	VAS scores to TS and CAS were recorded at baseline (T0) and immediately after application (T1), 1 week (T2), and after 1 (T3), 3 (T4) and 6 (T5) months	TTCP/DCPA showed a significant reduction in DH immediately and after 6 months.
**Porciani et al., ** **2014, Italy [46]**	100 (NR)	18–65 years 41.35 (NR)	RCT, DB	G1 (*n* = 50): calcium HA/DCPD; G1 (*n* = 50): Placebo. Two pieces of gum to chew together, three times per day, for 2 weeks	Schiff index value for TS, CAS and cold water at baseline (T0), and after 1 (T1) and 2 (T2) weeks	Chewing gum containing HA/DCPD had a statistically significant reduction in DH after one and two weeks.
**Vano et al., ** **2014, Italy [47]**	105 (NR)	20–70 years (NR)	RCT, DB	G1 (*n* = 35): 15% n-HA paste; G2 (*n* = 35): fluoride paste; G3 (*n* = 35): placebo. Toothbrush during 2 min twice a day for 4 weeks	VAS scores for TS, and CAS were evaluated at baseline (T0) and after 2 (T1) and 4 (T2) weeks	n-HA toothpaste significantly reduced DH between baseline and 4 weeks.
**Ghassemi et al., ** **2009, USA, Canada [48]**	208 (NR)	20–64 years 42.22 (NR)	RCT, DB	G1 (*n* = 106): Single phase ACP + 0.24% NaF paste; G2 (*n* = 102): Placebo (0.24% NaF). Toothbrush for 1 min twice a day for 8 weeks	VAS score of CAS was evaluated at baseline (T0), 4 weeks (T1) and 8 weeks (T2)	The toothpaste containing ACP showed a significant reduction in CAS VAS score compared to T0.
**Geiger et al., ** **2003, Israel [49]**	30 (NR)	NR	RCT, DB	G1 (*n* = 15): ACP; G2 (*n* = 15): Placebo 60 s application	VAS scores of TS and CAS were evaluated at baseline (T0), after one week (T1), after four weeks (T2) and after six months (T3)	ACP showed immediate relief of sensibility after application, with CAS and TS stimulation.

ST, Sensitive Teeth; NR, not reported; RCT, randomized clinical trial; DB, double blind; G, group; n-HA, nano-hydroxyapatite; DH, dentin hypersensitivity; TS, tactile sensitivity; CAS, cold air sensitivity; VAS, visual analog scale; SSS, Schiff sensitivity scale; TMD, Teethmate Desensitizer; TTCP, tetracalcium phosphate; DCPD, dicalcium phosphate dihydrate; DD, dentin desensitizer; BWE, Bite & White ExSense; HAP, hydroxyapatite; nZnMgHAP, nano-Zn-Mg-hydroxyapatite; nFAP, nano-fluoroapatite; fTCP, fluoride tricalcium phosphate; NRS, numbered rating scale; NaF, fluoride ions; VRS, verbal ratin scale; SrCl2, strontium chloride; GLU, Gluma Desensitizer Power Gel; MSC, MS Coat One F; NAN, NanoSeal; DCPA, dicalcium phosphate anhydrous; ACP, amorphus calcium phosphate; SCPS, calcium sodium phosphosilicate.

**Table 3 bioengineering-10-00447-t003:** Mean difference and standard deviations (SD) of visual analogic scale scores between 4 weeks follow-up and baseline after calcium phosphates application (subgroup analysis according to the test realized).

	Study Reference	Dentin Desensitizing Agents	Manufacturer	No.of Participants (Teeth)	Baseline Mean ± SD	4 Weeks Follow-up Mean ± SD	Mean Difference Random, 95% CI	*p*-Value
**Air blast test**	Poliakova et al., 2022, Russia [30]	Toothpaste (20% n−HAP)	NR	10	2.5 ± 0.53	1.3 ± 0.48	−1.20 ± 0.22 (−1.67 to −0.72)	<0.05
	Eyuboglu et al., 2020, Turkey [33]	Teethmate™ Desensitizer (TTCP/DCPA)	Kuraray Noritake Osaka, Japan	10 (21)	5.52 ± 1.66	2.14 ± 0.22	−3.38 ± 0.36 (−4.11 to −2.64)	<0.05
	Usai et al., 2019, Italy [34]	Teethmate™ Desensitizer (TTCP/DCPA)	Kuraray Noritake Dental Inc., Tokyo, Japan	35	4 ± 2.96	0.66 ± 1.51	−3.34 ± 0.56 (−4.46 to −2.21)	<0.05
	Usai et al., 2019, Italy [34]	Dentin Desensitizer (gel phase of n−HAP)	Ghimas, Casalecchio di Reno, Bologna, Italy	35	5.33 ± 2.22	1.66 ± 2.22	−3.67 ± 0.53 (−4.72 to −2.61)	<0.05
	Usai et al., 2019, Italy [34]	Bite&White ExSense (gel phase of n−HAP in a water)	Cavex Holland, Haarlem, Netherlands	35	4.33 ± 2.96	1.0 ± 0.01	−3.33 ± 0.50 (−4.32 to −2.33)	<0.05
	Ameen et al., 2018, Egypt [36]	15% nHAP	NR	(10)	3.01 ± 0.01	0 ± 0	−3.01 ± 0.00 (−3.00 to −2.99)	<0.05
	Ameen et al., 2018, Egypt [36]	15% nHAP + 1%NaF	NR	(10)	2.6 ± 0.52	0 ± 0	−2.59 ± 0.16 (−2.93 to −2.24)	<0.05
	Ameen et al., 2018, Egypt [36]	25% nHAP	NR	(10)	3.01 ± 0.01	0 ± 0	−3.01 ± 0.00 (−3.00 to −2.99)	<0.05
	Ameen et al., 2018, Egypt [36]	25%nHAP +1%NaF	NR	(10)	2.6 ± 0.52	0 ± 0	−2.59 ± 0.16 (−2.93 to −2.24)	<0.05
	Vano et al., 2017, Italy [38]	Bite&White ExSense (Gel, 15% n−HA)	Cavex Holland, Haarlem, Netherlands	35	2.97 ± 0.42	1.64 ± 0.43	−1.33 ± 0.10 (−1.53 to −1.12)	<0.05
	De Oliveira et al., 2016, Brazil [39]	Nano P^®^ (hydroxyapatite)	FGM Ltd.a, Brazil	(39)	6.23 ± 2.72	3.54 ± 3.72	−2.69 ± 0.73 (−4.15 to −1.22	<0.05
	Wang et al., 2016, Brazil [40]	Desensibilize Nano−P (20% hydroxyapatite)	FGM−Dentscare, Joinville, Brazil	(31)	7.04 ± 1.62	4.10 ± 3.50	−2.94 ± 0.69 (−4.32 to −1.55)	<0.05
	Wang et al., 2016, Brazil [40]	Desensibilize Nano−P (20% hydroxyapatite) + 10% HA	FGM−Dentscare, Joinville, Brazil	(22)	7.04 ± 1.62	4.48 ± 2.57	−2.56 ± 0.64 (−3.86 to −1.25)	<0.05
	Gopinath et al., 2015, India [41]	Aclaim^TM^ (Nano−HAP)	Group Pharmaceuticals, Bangalore, India	18	7.06 ± 1.55	5.39 ± 1.33	−1.67 ± 0.48 (−2.64 to −0.69)	<0.05
	Mehta et al., 2014, India [45]	Teethmate™ Desensitizer (TTCP/DCPA)	Kuraray Noritake Osaka, Japan	(50)	6.4 ± 0.5	2.2 ± 0.2	−4.20 ± 0.07 (−4.35 to −4.04)	<0.05
	Vano et al., 2014, Italy [47]	PrevDent^®^ toothpaste (15% n−HA)	NR	35	2.82 ± 0.35	1.2 ± 0.49	−1.62 ± 0.10 (−1.82 to −1.41)	<.05
	Ghassemi et al. 2009, USA, Canada [48]	Enamel Care (n−HAP)	NR	106	6.34 ± 1.11	3.47 ± 2.25	−2.87 ± 0.24 (−3.35 to −2.38)	<0.05
	Total *N*			319 (203)				
	Total mean score (SD)				4.63 ± 1.01	1.92 ± 1.28	−2.71 ± 0.07 (−2.85 to −2.57)	<0.05
**Tactile sensitivity test**	Eyuboglu et al., 2020, Turkey [33]	Teethmate Desensitizer (TTCP/DCPA)	Kuraray Noritake Osaka, Japan	10 (21)	2.85 ± 1.19	1.04 ± 1.20	−1.81 ± 0.36 (−2.55 to −1.06)	<0.05
	Usai et al., 2019, Italy [34]	Teethmate™ Desensitizer (TTCP/DCPA)	Kuraray Noritake Dental Inc., Tokyo, Japan	35	4.01 ± 2.96	0.66 ± 1.48	−3.35 ± 0.56 (−4.46 to −2.23)	<0.05
	Usai et al., 2019, Italy [34]	Dentin Desensitizer (gel phase of n−HAP)	Ghimas, Casalecchio di Reno, Bologna, Italy	35	5.33 ± 2.22	1.33 ± 2.22	−4.00 ± 0.53 (−5.05 to −2.94)	<0.05
	Usai et al., 2019, Italy [34]	Bite&White ExSense (gel phase of n−HAP in a water)	Cavex Holland, Haarlem, Netherlands	35	4.33 ± 2.96	0.01 ± 0.02	−4.32 ± 0.38 (−5.08 to −3.55)	<0.05
	Ameen et al., 2018, Egypt [36]	15% nHAP	NR	(10)	2.4 ± 1.03	0 ± 0	−2.39 ± 0.32 (−3.07 to −1.70)	<0.05
	Ameen et al., 2018, Egypt [36]	15% nHAP + 1%NaF	NR	(10)	2.2 ± 0.52	0 ± 0	−2.19 ± 0.16 (−2.53 to −1.84)	<0.05
	Ameen et al., 2018, Egypt [36]	25% nHAP	NR	(10)	2.8 ± 1.03	0 ± 0	−2.79 ± 0.32 (−3.47 to −2.10)	<0.05
	Ameen et al., 2018, Egypt [36]	25%nHAP +1%NaF	NR	(10)	2.3 ± 0.52	0 ± 0	−2.29 ± 0.16 (−2.63 to −1.94)	<0.05
	Vano et al., 2017, Italy [38]	Bite&White ExSense (Gel, 15% n−HA)	Cavex Bite&White ExSense, Cavex Holland BV	35	3.17 ± 0.49	1.83 ± 0.63	−1.34 ± 0.14 (−1.60 to −1.07)	<0.05
	Gopinath et al., 2015, India [41]	Aclaim^TM^ (Nano−HAP)	Group Pharmaceuticals, Bangalore, India	18	4.67 ± 1.08	3.78 ± 0.94	−0.89 ± 0.33 (−1.57 to −0.20)	>0.05
	Mehta et al., 2014, India [43]	Teethmate Desensitizer (TTCP/DCPA)	Kuraray Noritake Osaka, Japan	(50)	6.21 ± 1.82	2.81 ± 1.06	−3.4 ± 0.29 (−3.99 to −2.80)	<0.05
	Vano et al., 2014, Italy [47]	PrevDent^®^ toothpaste (15% n−HA)	NR	35	2.54 ± 0.52	0.95 ± 0.59	−1.59 ± 0.13 (−1.85 to −1.32)	<0.05
	Total *N*			158 (111)				
	Total mean score (SD)				3.56 ± 0.91	1.03 ± 0.72	−2.53 ± 0.07 (−2.66 to −2.39)	<0.05
**Cold water test**	Ameen et al., 2018, Egypt [36]	15% nHAP	NR	(10)	3 ± 0	0 ± 0	−2.99 ± 0.01 (−2.99 to −2.98)	<0.05
	Ameen et al., 2018,Egypt [36]	15% nHAP + 1%NaF	NR	(10)	2.6 ± 0.52	0 ± 0	−2.59 ± 0.16 (−2.93 to −2.24)	<0.05
	Ameen et al., 2018, Egypt [36]	25% nHAP	NR	(10)	3 ± 0	0 ± 0	−2.99 ± 0.01 (−2.99 to −2.98)	<0.05
	Ameen et al., 2018, Egypt [36]	25%nHAP + 1%NaF	NR	(10)	2.4 ± 0.52	0 ± 0	−2.39 ± 0.16 (−2.73 to −2.04)	<0.05
	De Oliveira et al., 2016, Brazil [39]	Nano P^®^ (hydroxyapatite)	FGM Ltd.a, Brazil	(39)	9.14 ± 1.37	6.51 ± 3.65	−2.63 ± 0.62 (−3.87 to −1.38)	<0.05
	Gopinath et al., 2015, India [41]	Aclaim^TM^ (Nano−HAP)	Group Pharmaceuticals, Bangalore, India	18	6.72 ± 1.01	4.94 ± 1.05	−1.78 ± 0.34 (−2.47 to −1.08)	<0.05
	Total *N*			18 (79)				
	Total mean score (SD)				4.47 ± 0.72	1.91 ± 1.46	−2.56 ± 0.16 (−2.88 to −2.23)	<0.05

A statistically significant decrease in the level of pain associated with dentin hypersensitivity was observed between 4 weeks of follow-up and baseline according to VAS score of air blast, tactile sensitivity, and cold water tests (*p* < 0.05).

## Data Availability

Not applicable.

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
