# Peer review of "The Effectiveness of Calcium Phosphates in the Treatment of Dentinal Hypersensitivity: A Systematic Review"

_bioengineering, 2023, doi:10.3390/bioengineering10040447_

Round 1
Reviewer 1 Report (Previous Reviewer 2)
The manuscript was revised properly and all the questions were addressed by the authors. It is now acceptable for publication.
Author Response
We thank you for your excellent review, which helped us to produce this new version. We are pleased that you find it acceptable for publication.
Reviewer 2 Report (Previous Reviewer 1)
The authors present a systematic review evaluating the efficacy of calcium phosphates as a treatment for dentin hypersensitivity.
Some English corrections are needed. See, for instance, lines 40, 63, 76, 107, and 125.
Lines 60-63: several SRs exist on this topic, and the results are sometimes contradictory. Also, it is important to detail the active principle being tested and not only its mechanism of action. Therefore, I suggest you increment this part with more information.
Your PICO question is wrongly formulated. First, your comparison is not applicable since you are not comparing calcium phosphate products with another product type, but instead, the before and after. Also, your outcome is pain reduction or hypersensitivity reduction, not “to determine the effectiveness”.
Search formulas are cut in the document and cannot be correctly evaluated.
Did you not use language filters in your literature search?
Please clearly define your inclusion and exclusion criteria. Your reasons for exclusion reported in figure 1 need to be detailed in your inclusion and exclusion criteria.
Section 2.1: for an article being reviewed in February 2022, a literature search until June 2022 is outdated. Please update your literature search.
Figure 1: what does “specialized reviex” means?
I don’t see the need for three tables instead of just one. In my opinion, they only difficult for the readers' comprehension. For example, in reading table 2, HD reduction values are not shown. Reading tables 3 and 4, the reader does not know how the products were administered. So, to get all the information, the readers must go to one table and the other.
Table 1: what ST (second column) means?
Discussion lines 18-23: this sentence is incorrect. Previous systematic reviews have evaluated hydroxyapatite, including its different forms of reducing HD. See, for instance, 10.1111/joor.12842.
Author Response
The authors present a systematic review evaluating the efficacy of calcium phosphates as a treatment for dentin hypersensitivity.
Dear Reviewer,
Thank you for the excellent review of our manuscript “The effectiveness of calcium phosphates in the treatment of dentinal hypersensitivity: a systematic review”. Your remarks and comments have allowed us to make the necessary corrections. We have corrected your remarks point by point. These modifications appear in the corrected version of the article. We hope that the revised version will meet your approval.
Some English corrections are needed. See, for instance, lines 40, 63, 76, 107, and 125.
Our response: we thank you for this comment. We had a proofreading done by a bilingual English speaker.
Lines 60-63: several SRs exist on this topic, and the results are sometimes contradictory. Also, it is important to detail the active principle being tested and not only its mechanism of action. Therefore, I suggest you increment this part with more information.
Our response: We thank you for this comment, whose answers have enabled us to improve the quality of the manuscript, changes were made from line 52 to 87. We added new references and updated the bibliography.
- Shiau, H.J. Dentin hypersensitivity. J Evid Based Dent Pract 2012, 12, 220-228, doi:10.1016/S1532-3382(12)70043-X.
- Magno, M.B.; Nascimento, G.C.R.; Da Penha, N.K.S.; Pessoa, O.F.; Loretto, S.C.; Maia, L.C. Difference in effectiveness between strontium acetate and arginine-based toothpastes to relieve dentin hypersensitivity. A systematic review. American journal of dentistry 2015, 28, 40-44.
- Marto, C.M.; Baptista Paula, A.; Nunes, T.; Pimenta, M.; Abrantes, A.M.; Pires, A.S.; Laranjo, M.; Coelho, A.; Donato, H.; Botelho, M.F.; et al. Evaluation of the efficacy of dentin hypersensitivity treatments-A systematic review and follow-up analysis. J Oral Rehabil 2019, 46, 952-990, doi:10.1111/joor.12842.
Your PICO question is wrongly formulated. First, your comparison is not applicable since you are not comparing calcium phosphate products with another product type, but instead, the before and after. Also, your outcome is pain reduction or hypersensitivity reduction, not “to determine the effectiveness”.
Our response: Thank you for this valuable suggestion. Indeed, the aim of our study is not to compare calcium phosphates with other types of products used in DH, but rather to see if it has an effect on the level of DH pain before and after treatment. We modified the text according to your comment at line 138 to 139.
Search formulas are cut in the document and cannot be correctly evaluated.
Our response: Thank you for pointing out this layout error. We corrected it.
Did you not use language filters in your literature search?
Our response: In accordance with PRISMA 2020 guidelines, no linguistic criteria have been applied to allow for a more comprehensive search
Please clearly define your inclusion and exclusion criteria. Your reasons for exclusion reported in figure 1 need to be detailed in your inclusion and exclusion criteria.
Our response: Thank you for pointing out this oversight, we have detailed the selection criteria as indicated in the commentary line 142 to 146.
Section 2.1: for an article being reviewed in February 2022, a literature search until June 2022 is outdated. Please update your literature search.
Our response: you are completely right, this is a mistake. Indeed, following the previous comments of the reviewers, we were asked to better define the search equation. A new bibliographic search was carried out in December 2022 and allowed us to identify 5 more references. The dates have been changed in the text line 149. Here the references of the studies mentioned:
- Amaechi, B.T.; Lemke, K.C.; Saha, S.; Luong, M.N.; Gelfond, J. Clinical efficacy of nanohydroxyapatite-containing toothpaste at relieving dentin hypersensitivity: an 8 weeks randomized control trial. BDJ Open 2021, 7, 23, doi:10.1038/s41405-021-00080-7.
- Usai, P.; Campanella, V.; Sotgiu, G.; Spano, G.; Pinna, R.; Eramo, S.; Saderi, L.; Garcia-Godoy, F.; Derchi, G.; Mastandrea, G.; et al. Effectiveness of Calcium Phosphate Desensitising Agents in Dental Hypersensitivity Over 24 Weeks of Clinical Evaluation. Nanomaterials (Basel) 2019, 9, doi:10.3390/nano9121748.
- Amaechi, B.T.; Lemke, K.C.; Saha, S.; Gelfond, J. Clinical Efficacy in Relieving Dentin Hypersensitivity of Nanohydroxyapatite-containing Cream: A Randomized Controlled Trial. Open Dent J 2018, 12, 572-585, doi:10.2174/1874210601812010572.
- Ameen, S.; Niazy, M.; El-yassaky, M.; Jamil, W.; Attia, M. Clinical Evaluation of Nano-Hydroxyapatite as Dentin Desensitizer. Al-Azhar Dental Journal for Girls 2018, 5, 79-87.
- Naoum, S.J.; Lenard, A.; Martin, F.E.; Ellakwa, A. Enhancing Fluoride Mediated Dentine Sensitivity Relief through Functionalised Tricalcium Phosphate Activity. Int Sch Res Notices 2015, 2015, 905019, doi:10.1155/2015/905019.
Figure 1: what does “specialized reviex” means?
Our response: The error has been corrected
I don’t see the need for three tables instead of just one. In my opinion, they only difficult for the readers' comprehension. For example, in reading table 2, HD reduction values are not shown. Reading tables 3 and 4, the reader does not know how the products were administered. So, to get all the information, the readers must go to one table and the other.
Our response: Thank you for this comment which is very relevant. Indeed, as you have noticed, Table 2 contains only the summary of qualitative information on the different studies selected. Adding quantitative data to this table (2) would make it very voluminous and will undoubtedly drown out the information related to our study objective. In order to highlight the level of pain reduction following treatment with calcium phosphate between baseline and 4 weeks of follow-up, we considered it relevant to perform subgroup analyses according to the clinical test used by the authors. This pooling of data allowed us to observe a significant reduction in the level of pain regardless of the test. Also, our approach is explained by the fact that some of the included studies do not present follow-up data at 4 weeks.
For simplicity, we have combined Tables 3, 4 and 5 into a single table, noting that they are subgroups. Although we did not generate a Forest plot, the summary of the data is similar to that of Melo Alancar C et al. Here is the reference of the study in question.
- de Melo Alencar, C.; de Paula, B.L.F.; Guanipa Ortiz, M.I.; Barauna Magno, M.; Martins Silva, C.; Cople Maia, L. Clinical efficacy of nano-hydroxyapatite in dentin hypersensitivity: A systematic review and meta-analysis. J Dent 2019, 82, 11-21, doi:10.1016/j.jdent.2018.12.014.
Table 1: what ST (second column) means?
Our response: Thank you for your comment. This is the number of sensitive teeth assessed in each study, we have added the meaning of the acronym in the table legend.
Discussion lines 18-23: this sentence is incorrect. Previous systematic reviews have evaluated hydroxyapatite, including its different forms of reducing HD. See, for instance, 10.1111/joor.12842.
Our response: Thank you for this valuable suggestion, we totally agree with this comment. We had not identified this interesting systematic review and meta-analysis. We have amended the discussion accordingly on line 18 and lines 33 to 37. We have also updated the bibliography.
Round 2
Reviewer 2 Report (Previous Reviewer 1)
The authors present a systematic review evaluating the efficacy of calcium phosphates as a treatment for dentin hypersensitivity.
Unfortunately, your work lacks coherence. In the abstract section (lines 15-17), you state your inclusion criteria are “clinical randomized controlled studies using calcium phosphates in treating dentin hypersensitivity, compared with placebo or usual treatment”. In your PICO question (117-119), you state are not comparing with other treatments or placebo, but instead the before and after of calcium phosphates administration. Then, in your result tables, you are again comparing with other treatments. Both options are valid. You can choose to perform an SR to compare to other treatments or do an SR to evaluate the efficacy of just calcium phosphates before and after. But you have to choose an option and present a coherent work.
PubMed search formula: your formula is incorrect. You can not use MeSH terms and add * to them. The MeSH terms must be searched precisely how they are inserted in the database. Also, “dentine hypersensitivity” is not the correct MeSH term, and you failed to include the necessary synonyms. As I previously suggested, I recommend you ask for the help of a librarian or a colleague with expertise in literature search to help you with your search formulas. Without this, it is impossible to perform a correct SR.
PubMed search: since you only wanted to include RCT, why did you not use the “randomized controlled trial” filter instead of “clinical trial”?
Figure 1: for each reason on “Reports excluded based on exclusion criteria,” you must indicate the number of studies.
Author Response
Dear Reviewer,
Thank you again for your excellent review of our manuscript “The effectiveness of calcium phosphates in the treatment of dentinal hypersensitivity: a systematic review”. Your remarks and comments have allowed us to make the necessary corrections. We have corrected your remarks point by point. These modifications appear in the corrected version of the article. We hope that the revised version will meet your approval.
Unfortunately, your work lacks coherence. In the abstract section (lines 15-17), you state your inclusion criteria are “clinical randomized controlled studies using calcium phosphates in treating dentin hypersensitivity, compared with placebo or usual treatment”. In your PICO question (117-119), you state are not comparing with other treatments or placebo, but instead the before and after of calcium phosphates administration. Then, in your result tables, you are again comparing with other treatments. Both options are valid. You can choose to perform an SR to compare to other treatments or do an SR to evaluate the efficacy of just calcium phosphates before and after. But you have to choose an option and present a coherent work.
Our response: Thank you for your comment, we have indeed removed all notions of comparison with other products still present in the summary and in table 2
PubMed search formula: your formula is incorrect. You can not use MeSH terms and add * to them. The MeSH terms must be searched precisely how they are inserted in the database. Also, “dentine hypersensitivity” is not the correct MeSH term, and you failed to include the necessary synonyms. As I previously suggested, I recommend you ask for the help of a librarian or a colleague with expertise in a literature search to help you with your search formulas. Without this, it is impossible to perform a correct SR. PubMed search: since you only wanted to include RCT, why did you not use the “randomized controlled trial” filter instead of “clinical trial”?
Our response: Thank you for pointing out this error. We have generated a new pubmed search equation, and updated the flow chart and the text. For this new search equation, we have applied the Randomized Control Trial filter.
Figure 1: for each reason on “Reports excluded based on exclusion criteria,” you must indicate the number of studies.
Our response: Thank you for your attention, we have added the number of items excluded for each of various reasons listed.
This manuscript is a resubmission of an earlier submission. The following is a list of the peer review reports and author responses from that submission.
Round 1
Reviewer 1 Report
The authors present a systematic review evaluating the efficacy of calcium phosphates as a treatment for dentin hypersensitivity.
Some English corrections are needed. See, for instance, lines 181 and 207.
Lines 13-15 and 101-105: please provide a more specific aim. If you are performing a systematic review, you aim to answer a specific question (in this manuscript, if calcium phosphates are effective in reducing HD or not) and not to explore or evidence something.
Lines 18-19: publication language and date are not inclusion criteria but filters you use during your literature search. Please correct.
Line 20: please indicate the databases that were searched.
I am confused regarding the materials you intended to evaluate. In lines 96-97, you refer to “no systematic study ever focused on the effectiveness of all calcium phosphate materials which are able to self-set to a hard-mass, such as tetracalcium phosphate (TTCP) and dicalcium phosphate dihydrate (DCPD”. Later on, in lines 101-103 you refer to “various calcium phosphate (nano-hydroxyapatite, TCP, DCPA and ACP”. Why did you exclude hydroxyapatite? If you intended only to evaluate self-set materials, why did you include nano-hydroxyapatite? Please clarify. Also, in the manuscript results and discussion, you refer several times to calcium phosphates which includes hydroxyapatite.
Why did you limit your search from January 2000 to June 2022?
Why did you limit your search to the English language? For instance, why did you not include articles published in French? Potentially this could have allowed you to collect some more articles.
Following PRISMA 2020 guidelines, please provide the search formulas to all databases to allow a proper evaluation of your literature search.
Your PubMed search formula is incorrect. You need to use the appropriate MeSH terms and their synonyms and to correct the brackets. Also, using some search words is unnecessary and limits the articles you could have obtained that are relevant for your review. I recommend you search for help from a librarian or a colleague with experience in literature search.
Line 164: you refer to the inclusion of 3413 records, while figure 1 refers to 3410. What is the correct number?
Line 177: I suggest removing this sentence. Since your inclusion criteria were RCTs, all the included studies were RCTs.
The results section mainly describes the studies characteristics and not the results regarding the products´ efficacy, active ingredient… (which are only presented in table 1). Please correct it. Consider reducing section 3.2. For instance, I find the description of the stimulus unnecessary.
Please place figure 1 and table 1 near the correspondent text.
Figure 1: please use the appropriate PRISMA 2020 flow diagram.
Table 1: please add a table caption describing the abbreviations' meaning.
Table 1, second column. Since this column refers to the number of participants or number of teeth, when only one n is presented, it is referring to what?
Table 1: I suggest you indicate the number of participants/teeth per group.
Table 1: the studies included do not present quantitative data? For instance, the percentage of DH reduction?
Discussion section: the authors failed to include other relevant articles, namely systematic reviews on this topic. Please update your references.
Reviewer 2 Report
This systematic review aimed to analyze studies on the use of calcium phosphates for dentin hypersensitivity treatment.
The methods are well described. However, the discussion section needs deep improvement, including a critical analyzes of the studies. Conclusion should be revised, considering the studies analyzed in this systematic review.
Abstract – Clinical significance.
In “This is a significant challenge for clinicians as current treatments are not always sustainable.” – Please consider revising, it is not clear what authors mean by sustainable.
In “Calcium phosphates seem to be a promising treatment, which must be more evaluated over longer periods of time.”. Please, include a more assertive statement. Based on this systematic review, what could be effective concluded?
Why do the authors stated that “the treatment must be more evaluated over longer periods of time”?
Line 268 – Please discuss on the cause of dentin hypersensitivity in the studied populations. These studies included patients from 18 to 70 yrs (wide range). How different etiologies could have influenced the observations? Include information in summary Table.
Line 270 – Were these statements based on the studies included in the review? Detail and discuss, including the references.
Line 276 – Include a discussion on the other products or placebo used in the evaluated studies. Were they comparable? What were the most commonly used controls?
Line 276 – Include reference for “The fact that ten of the fifteen selected studies were classified as having a « low risk of bias makes this systematic review reliable.”
Line 279 – This paragraph seems to be out of context, please revise.
Line 279 – The period “Our systematic review reinforces recent data demonstrating an increasing interest in calcium phosphate, and more precisely hydroxyapatite, in the treatment of remineralization of incipient carious lesions or management of intra-oral biofilm” seems to be out of context. Please, consider revising.
Line 294 – This period seems to be unfinished, please revise.
Line 316- This period “A level of pain still persisted and the effectiveness of the treatment tend to decrease or disappeared within a few weeks” should be revised.
Conclusion
Make clearer the evidences that support the conclusion “This systematic review shows a reduction in pain perception after calcium phosphate application”
Reviewer 3 Report
Dear Editor and Authors,
The present systematic review manuscript is very interesting and the topic is within the aims of the Journal.
However, for the organization and for the large information, understanding the manuscript is very difficult. Please re-organize. Thanks.